# Deep causal speech enhancement and recognition using efficient long-short term memory Recurrent Neural Network

**Zhenqing Li**[1], **Abdul Basit**[ID][1]*, **Amil Daraz**[ID][1], **Atif Jan**[2]

**1** School of Information Science and Engineering, NingboTech University, Ningbo, China, **2** Department of Electrical Engineering, University of Engineering and Technology, Peshawar, Pakistan

* abdulbasit@nbt.edu.cn

**Data Availability Statement:** All relevant data are within the paper.

**Funding:** This work is supported by" Talent Introduction Fund Project of Ningbo Tech University under grant no 20211009. The funder Dr

## Abstract

Long short-term memory (LSTM) has been effectively used to represent sequential data in recent years. However, LSTM still struggles with capturing the long-term temporal dependencies. In this paper, we propose an hourglass-shaped LSTM that is able to capture long-term temporal correlations by reducing the feature resolutions without data loss. We have used skip connections in non-adjacent layers to avoid gradient decay. In addition, an attention process is incorporated into skip connections to emphasize the essential spectral features and spectral regions. The proposed LSTM model is applied to speech enhancement and recognition applications. The proposed LSTM model uses no future information, resulting in a causal system suitable for real-time processing. The combined spectral feature sets are used to train the LSTM model for improved performance. Using the proposed model, the ideal ratio mask (IRM) is estimated as a training objective. The experimental evaluations using short-time objective intelligibility (STOI) and perceptual evaluation of speech quality (PESQ) have demonstrated that the proposed model with robust feature representation obtained higher speech intelligibility and perceptual quality. With the TIMIT, LibriSpeech, and VoiceBank datasets, the proposed model improved STOI by 16.21%, 16.41%, and 18.33% over noisy speech, whereas PESQ is improved by 31.1%, 32.9%, and 32%. In seen and unseen noisy situations, the proposed model outperformed existing deep neural networks (DNNs), including baseline LSTM, feedforward neural network (FDNN), convolutional neural network (CNN), and generative adversarial network (GAN). With the Kaldi toolkit for automated speech recognition (ASR), the proposed model significantly reduced the word error rates (WERs) and reached an average WER of 15.13% in noisy backgrounds.

## Introduction

Speech enhancement is a signal processing technique that aims to improve the quality and intelligibility of speech signals that are degraded by various types of noise, such as background noise, reverberation, and channel distortions. In practice, speech enhancement techniques typically operate in the time-frequency domain, where the speech signal is represented as a sequence of short-time Fourier transforms (STFTs). By analyzing the speech signal in the

Li is the main author of this work and he contributed fully to this work in the way of simulations and original paper writing. He is the project leader of the "Talent Introduction Fund Project of Ningbo Tech University under grant no 20211009."

**Competing interests:** The authors have declared that no competing interests exist.

frequency domain, it is possible to identify and isolate the components that are corrupted by noise, while preserving the underlying speech components. Speech enhancement is a crucial component in many applications, such as hearing aids, telecommunication systems, and speech recognition systems. By improving the quality and intelligibility of speech signals, speech enhancement techniques can significantly enhance the performance and usability of these systems. There are many different signal processing techniques that can be used for speech enhancement, such as spectral subtraction [1], Wiener filtering [2], and non-negative matrix factorization (NMF) [3]. These techniques aim to reduce or remove the noise component from the speech signal while preserving the speech content. One of the key challenges in speech enhancement is to distinguish between the desired speech signal and the noise component. This is particularly challenging in the presence of non-stationary noise, which can vary in both time and frequency domains. To overcome this challenge, speech enhancement systems often use adaptive algorithms that can track the changes in the noise statistics and adjust the filtering parameters accordingly. The performance of speech enhancement systems is typically evaluated using objective measures, such as signal-to-noise ratio (SNR) and perceptual evaluation of speech quality (PESQ), as well as subjective listening tests. In general, speech enhancement can significantly improve speech signals' perceived quality and intelligibility, particularly in noisy environments.

Deep learning techniques have shown a lot of promise in improving speech enhancement performance in non-stationary noisy environments, where the characteristics of the noise may change over time [4–6], and show its effectiveness in other applications [7–10]. Deep neural networks (DNNs) are effective models for speech enhancement because they can learn the nonlinear relationship between input and output features. In particular, deep learning-based speech enhancement models, such as Convolutional Neural Networks (CNNs), Recurrent Neural Networks (RNNs), and Deep Neural Networks (DNNs), can learn to extract features that are robust to various types of noise and can adapt to changing noise conditions over time. Deep learning-based speech enhancement models have shown significant improvements in speech quality and intelligibility, particularly in challenging environments, such as noisy speech in cars, on cell phones, or in crowded public places. There are two main types of DNN-based speech enhancement algorithms: masking-based [11, 12] and mapping-based [13–15]. Masking-based algorithms have been found to be more effective because they can estimate time-frequency (T-F) masks as training targets, which can better track the target speaker and produce better de-noising results. Fully connected feedforward DNNs (FDNNs) have been commonly used in speech enhancement, but they are limited by short context windows and cannot capture long-term context information. Multi-layer networks are used in DNN-based speech enhancement methods to overcome this limitation and provide better performance in non-stationary noisy environments. Overall, DNN-based speech enhancement techniques can provide superior de-noising results without requiring statistical features or distribution assumptions. However, they require large amounts of training data and computational resources, which can be a limitation in some applications.

Recurrent Neural Networks (RNNs) are a type of neural network that can process sequential data and capture long-range temporal dependencies. They are particularly well-suited for natural language processing (NLP) tasks that involve variable-length sequences of data, such as speech waveforms, text, and time series. RNNs have also been successfully used for other NLP tasks, such as speech recognition and dialogue modeling. For example, in speech recognition, RNNs can be used to model the relationship between an input speech waveform and its corresponding text transcription. According to research [16, 17], it is preferable to structure speech enhancement as a sequence-to-sequence process in order to regulate long-term context windows. RNNs [18], CNNs [19], and GANs [20] have been presented where networks are trained

and evaluated with various noise types and speakers of both genders. The authors propose a four-hidden-layer LSTM model for speaker generalization [16]. Regarding speech intelligibility, the findings demonstrated that the LSTM model generalized better to untrained speakers and significantly outperformed a DNN-based model. Numerous studies demonstrate that with sequence-to-sequence processing, LSTM may successfully manage long-term context windows and be effective in SE [21, 22]. The difficulty of capturing long-term dependencies is a crucial obstacle RNN models face when attempting to model extended sequences of input data. In addition, training RNNs via Back Propagation Through Time (BPTT) exposes gradients to vanishing and explosion. LSTM [23, 24] and gated recurrent unit (GRU) [25, 26] are examples of RNN variations that use unique transition functional units and optimization strategies to address these difficulties. Layered RNNs [27] and skip RNNs are two of the existing focused architectures [16]. A causal dynamic model using attention LSTM encoder-decoder is proposed for SE with excellent noise reduction and speech recognition results- [28]. A time-domain brain-assisted speech enhancement model incorporates electroencephalography signals to extract the target speaker from monaural speech mixtures. The proposed SE model is based on the fully convolutional time-domain network [29]. Another study [30] proposes a cooperative attention-based speech enhancement model and combines local and non-local attention operations in a learnable and self-adaptive manner. The study [31] proposes a multiscale attention metric generative adversarial network to avoid the mismatch between the objective function used to train the speech enhancement models and introduces the attention mechanism in the metric discriminator. Another study uses a Convolutional attention transformer bottleneck in the encoder-decoder framework for speech enhancement and obtains better SE and automatic speech recognition results [32].

In this paper, we describe LSTM models that are capable of capturing long-term temporal correlations and avoiding gradient decay across layers. The significant contributions of this study are emphasized as follows. (i) It is suggested that an hourglass-shaped LSTM model can capture long-term temporal sequence-to-sequence data and decrease feature resolutions without data loss in layers. (ii) In order to avoid gradient decay in nonadjacent layers, skip connections are introduced. (iii) In the skip connections, an attention gate is utilized to suppress irrelevant input and emphasize the critical spectral regions of features. (iv) Combined feature sets are extracted from the noisy speech to train LSTM models reliably. (v) IRM is estimated to be the training target for suppressing the additive noise from the target speech in order to obtain higher-quality and more intelligible speech.

The remainder of this paper is organized as follows. The proposed speech enhancement system is explained in Section 2. Experiments and setups are presented in Section 3. Results and discussions are presented in Section 4. Finally, conclusions are drawn in Section 5.

## Proposed speech enhancement

### Problem formulation

Consider that a clean speech signal $s(t)$ is deteriorated by additive background noise $d(t)$ and that the resultant noisy speech $y(t)$. Using the short-time Fourier Transform (STFT), the noisy speech $y(t)$ is transformed into the frequency domain, yielding the frequency-domain representation of $y(t)$ as $|Y(f, t)|$, where $t$ represents the frame index and $f$ represents the frequency index. A combined set of acoustic features is extracted to train the LSTM model reliably. The learned parameters estimate the time-frequency mask (IRM) as a training target during the testing phase. The calculated magnitude mask $|M(t, f)|$ is then multiplied by the magnitude of the noisy speech $|Y(t, f)|$ to reduce background noise signals in the underlying clean speech $|S(t, f)|$. During waveform reconstruction, the predicted magnitude and the noisy phase are

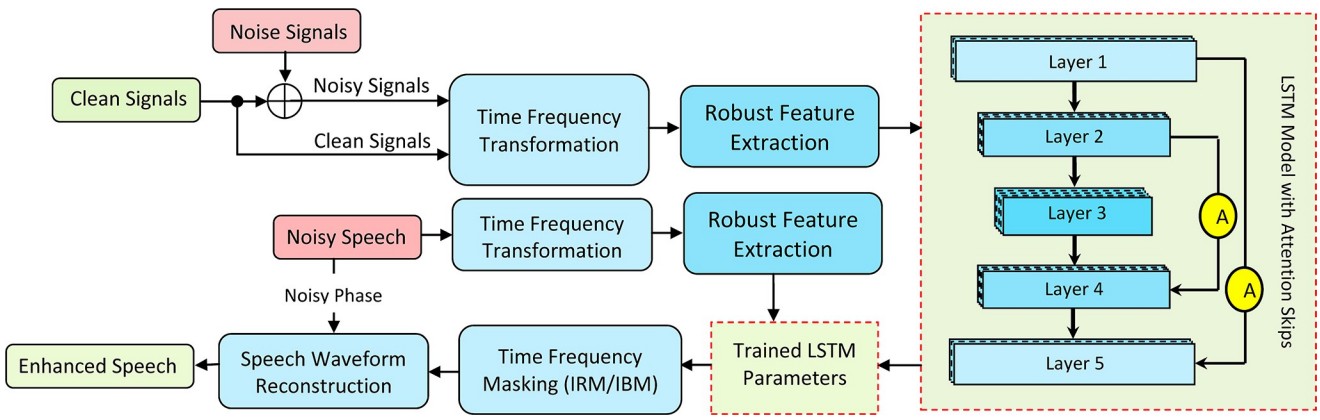

**Fig 1. The proposed speech enhancement.**

combined to generate improved speech. Fig 1 depicts the block diagram of the proposed speech enhancement.

## Proposed LSTM architecture

LSTMs are capable of capturing information from speech waveforms, which are essentially long-term temporal sequences. Using the following novel approach, the network has successfully circumvented the RNN's constraints. This suggested LSTM model is influenced by research by Abdulbaqi [33]. LSTM layers are first organized using an hourglass-shaped design. For the top pyramid (first two layers to the third layer), the number of time steps decreases as the number of neurons increases. Similarly, in the bottom pyramid (third to final two levels), the time steps are increasing while the number of neurons is decreasing. Instead of the typical LSTM's fixed neurons and time steps, we have employed an alternative technique to produce a compact and effective model for speech enhancement. The outputs of the model have been modified to favor fewer time steps. Reshaping the layer output to lower and increase the time steps eliminates data loss and enables the model to have a suitable number of neurons. With these architectural modifications, the model can manage high-resolution features without exceeding memory capacity and with fewer network parameters. Second, skip connections are used between pyramid layers of similar shape from the top pyramid to the bottom pyramid. Thus, the decreasing gradient across layers is maximized. Thirdly, the attention gate is used in skips to emphasize significant spectral areas. The speech spectrum contains formants with a sparse distribution in high-frequency regions and a predominance in low-frequency regions. Consequently, it is essential to differentiate the various spectral areas with varying weights using an attention gate. Fig 2 depicts the network's five LSTM layers and two attention skip connections. Table 1 expresses the time steps and units. The network finds the nonlinear relationship and converts the noisy speech signal $y(t)$ into a clean speech signal. Using the following equations where the forget gate is important.

$$i_t = \sigma(\mathbf{W}_i) \times [C_{t-1}, h_{t-1}, x_t] + b_i \tag{1}$$

$$f_t = \sigma(\mathbf{W}_f) \times [C_{t-1}, h_{t-1}, x_t] + b_f \tag{2}$$

$$O_t = \sigma\mathbf{W}_o \times [C_{t-1}, h_{t-1}, x_t] + b_o) \tag{3}$$

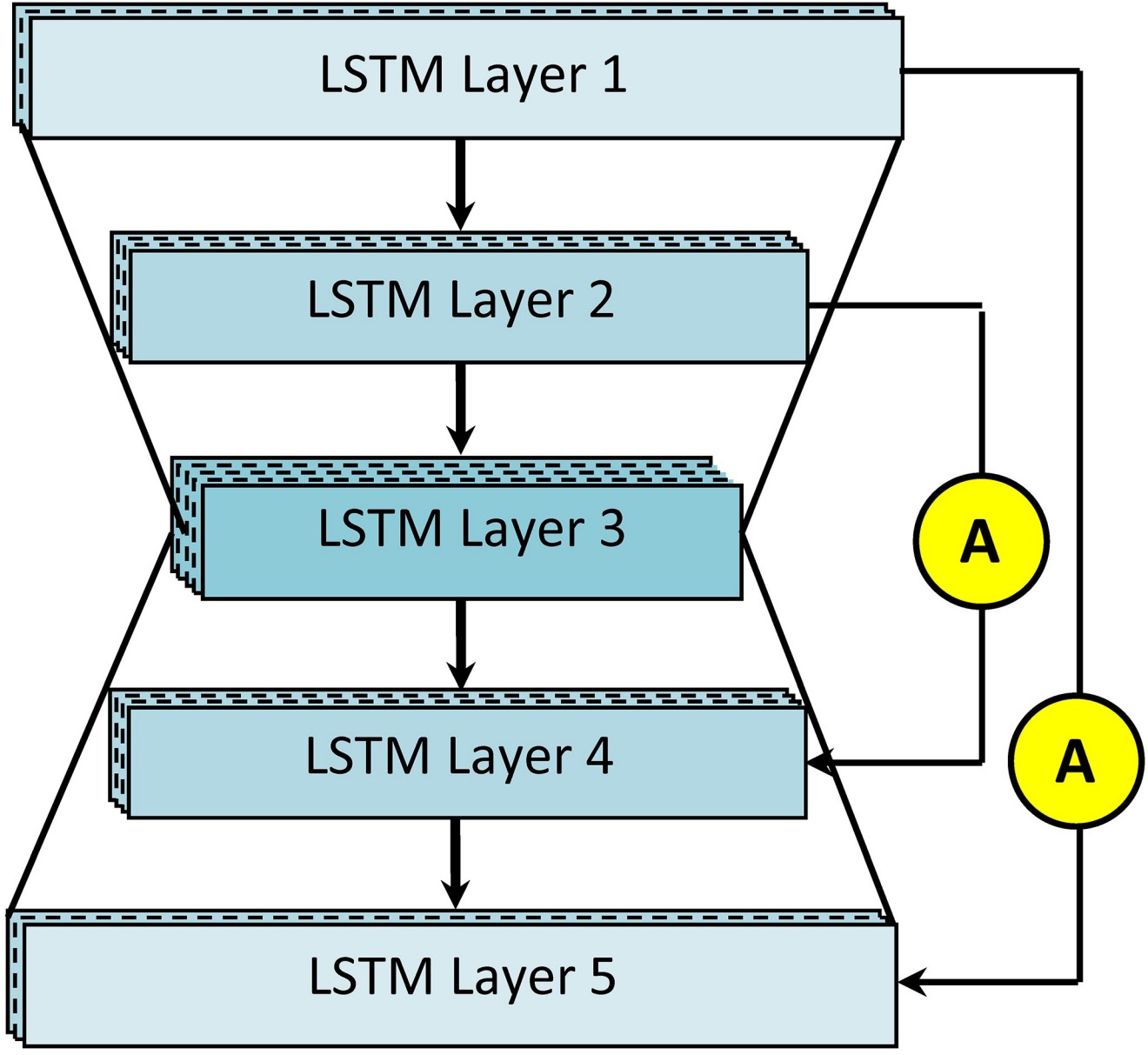

**Fig 2. The proposed LSTM Architecture.**

**Table 1. Proposed LSTM details.**

| Layer | Units in Layer | Time-Steps in Layer |
|---|---|---|
| 1 | 256 | 512 |
| 2 | 512 | 256 |
| 3 | 1014 | 128 |
| 4 | 512 | 256 |
| 5 | 256 | 512 |

$$k_t = tanh(\mathbf{W}_c \times [C_{t-1}, h_{t-1}, x_t] + b_c) \tag{4}$$

$$C_t = f_t \otimes C_{t-1} + i_t \otimes +k_t \tag{5}$$

where $W_i$, $W_f$, $W_o$, are weight matrices of input, forget, and output gate associated with hidden states, $x_t$ is input to the current timestamp, $h_{t-1}$ is hidden state of the previous timestamp, $C_{t-1}$ and $C_t$ shows the previous and current timestamp respectively whereas $b_i$, $b_f$, and $b_o$ are the biased terms of input, forget, and output gate, respectively. In the architecture, LSTMs are favored over RNNs because of their gated structure, superior training, and superior SE performance. This compact LSTM design enhances network capacity by sharing the hidden states across the similar and bottom layers. The lowering time-steps and increasing units (from upper layers to the middle layer) and increasing time-steps and decreasing units (from the middle layer to the bottom layer) allow for a more accurate portrayal.

The LSTM layers share their hidden states, hence the hidden states of an LSTM unit in layer l at time t are obtained by concatenating its hidden states, which are dependent on the lower layer l–1 at time t and this layer at time t–1. Before the skips, the hidden states of the top and lower layers are merged to form a final output with the same size as the input vectors.

$$h_t^l = LSTM(h_t^{l-1}, h_{t-1}^l) \tag{6}$$

The output will be created by combining the hidden states of all layers as:

$$X = LSTM(h_l^5, h_T^5) \tag{7}$$

where $X$ indicates the output from the last layer while $h_T^5$ indicates hidden states of last layer in the architecture. To avoid gradient decaying over the layers, two skips are added. The skips provide deep training and effective generalization after the combination of low-level features with high-level features. Speech spectra include different frequency components; the formants are usually dominant in the low-frequency regions and demonstrate a sparse distribution in the high-frequency regions. Hence, it is important to distinguish different spectral regions with different weights by using an attention process. Moreover, important regions and features are focused on improving the quality of output.

## Features combination

At the frame level, the feature sets are obtained from the speech signals. The frame shift and lengths were set at 10 and 20 milliseconds, respectively. These feature sets are comprised of 31-dimension Mel-Frequency Cepstral Coefficients (MFCC), 64-dimension Gammatone Filter-bank Energies (GFE), 15-dimension Amplitude Modulation Spectrogram (AMS), and 13-dimensions relative Spectral Transformed Perceptual Linear Prediction Coefficients (RASTA-PLP), given as:

$$f_S = f_s^{MFCC} + f_s^{RASTA-PLP} + f_s^{AMS} + f_s^{GFE} \tag{8}$$

$$f_Y = f_y^{MFCC} + f_y^{RASTA-PLP} + f_y^{AMS} + f_y^{GFE} \tag{9}$$

Here $d$ denotes the dimensions of features, $f_S$ and $f_Y$ are the combined feature vectors of clean and noisy speech. The gamma tone filterbank energies features are derived from the Cochleagrams, which is a T-F representation often employed in computational auditory scene analysis (CASA). It explains the operation of the human auditory system. A filter bank of 64 channels is used to generate the Cochleagrams. The delta features are also calculated and

**Table 2. Brief comparison in terms of features, training objective, DNN type, and loss function.**

| Model | Features | Training Objective | DNN Type | Loss Function |
|---|---|---|---|---|
| MO-LSTM [18] | Spectral Features | T-F Masking | LSTM+Phase | MSE Loss |
| 1D-CNN [19] | Waveform | Waveform | 1D-CNN | Phase+MSE |
| μ-law SGAN [20] | Compressed Spectrum | T-F Mask | GAN | MSE Loss |
| DeepResGRU [25] | Spectral Features | T-F Masking | Residual GRUs | MSE Loss |
| CS-DWL [28] | Spectral | T-F Masking | Attention LSTMs | Dynamical MSE |
| CASE-Net [30] | Complex Spectrum | T-F Masking | 2D+1D CNN | SI-SDR |
| MAMGAN [31] | Waveform Features | Waveform | GAN | MSE+SI-SNR |
| NSE-CATNet [32] | Spectral Features | T-F Mask | Conv-2D+Transformer | MSE Loss |
| CleanUNet [50] | Waveform Features | Waveform | Conv-1D+Self Attention | L1+M-STFT |
| UFLSTM [51] | Spectral Features | T-F Masking | Type-2 Fuzzy LSTM | MSE Loss |
| Proposed [–] | Spectral Features | T-F Masking | U-shaped-LSTM | MSE Loss |

attached to the features. Table 2 briefly compares the models in terms of features, training objective, DNN type, and loss function.

## Experiments

### Datasets

Various tests are undertaken using speech sentences selected from the TIMIT [34], LibriSpeech [35], and VoiceBank [36] to evaluate the performance of SE. LibriSpeech comprises 1000 hours of speech data at a 16 kHz sampling rate. The TIMIT also contains phonetically balanced speech data at a sampling rate of 16 kHz. The Voice Bank is composed of male and female speakers of the English language. In our research, only clean speech samples from databases were utilized. The Aurora-4 database [37], NOISEX-92 database [38], and DEMAND database [39] are selected to obtain background noises for evaluating the proposed speech enhancement methods. Four input SNRs (-8 dB, -4 dB, 0 dB, and 4 dB) are utilized to create noisy sentences. To train the proposed LSTM network, sentences from VoiceBank, TIMIT, and LibriSpeech are used in order to estimate the T-F mask. For a more accurate generalization of the speaker, the training sentences include male and female speakers combined with all noise sources. Consequently, a large quantity of speech sentences is selected for model training. In addition, a separate set of speech sentences is prepared at random from three databases (TIMIT, LibriSpeech, and VoiceBank) for model testing. Only two noise sources are excluded from training and these noises are termed unseen noises(factory2 and café).

### Network setting

In this article, a five-layered LSTM network is used where the input layer has a size of 1230 dimensions using the context windows of 11 frames. Every layer of the LSTM is comprised of N units and M time steps, while the output layer consists of 257 units. The BPTT (Backpropagation through time) is employed during training. Optimization is performed using adaptive gradient descent with momentum. There are 512 samples in each batch. During processing, the AGD scaling factor is fixed at 0.0010 whereas the learning rate is reduced linearly from 0.06 to 0.002. There are 80 epochs in all. is set at 0.4 for the first epochs, then momentum is raised to 0.8 for subsequent epochs. With a dropout rate of 0.2, dropout regularisation is implemented. During mask estimation, the MSE loss function is applied. The LSTM models do not employ future information, which is equivalent to causal systems. 11 frames of features

**Table 3. The details of the network with hyperparameters of the LSTM and competing deep learning models.**

| Hyperparameters | Baseline LSTM | DNN | Proposed LSTM |
|---|---|---|---|
| No of hidden layers | 5 | 5 | 5 |
| Layer 1 Units | 1024 | 1024 | 256 |
| Layer 2 Units | 1024 | 1024 | 512 |
| Layer 3 Units | 1024 | 1024 | 1014 |
| Layer 4 Units | 1024 | 1024 | 512 |
| Layer 5 Units | 1024 | 1024 | 256 |
| Learning Rate | 0.001 | 0.001 | 0.001 |
| No of Epochs | 80 | 80 | 80 |
| Momentum Rate | 0.8 | 0.8 | 0.8 |
| Dropout Rate | 0.2 | 0.2 | 0.2 |
| Loss Function | MSE | MSE | MSE |
| Activation | – | ReLU | – |

are concatenated as the network input at each time step. The input to the model is causal. However, as demonstrated in Table 1, the network's computing process varies. There are different time steps in different layers, the calculation of the first time step of the second layer requires the output of the second time step of the first layer, and the calculation of the first time step of the third layer requires the output of the second, third, and fourth time steps of the first layer; therefore, when calculating the first time step of the output layer, the future time step of the first layer must be used. The deep model hyperparameters are listed in Table 3. Here Units indicate the neurons.

## Evaluation metrics

Experiments use two objective metrics to objectively assess the proposed speech enhancement method. STOI (short-time objective intelligibility) and PESQ (perceptual evaluation of speech quality) determine the intelligibility and quality, respectively. ITU-T P.862 guideline PESQ [40] assesses the perceptual quality of noisy speech (between -0.5 to 4.5). STOI [41] assesses the intelligibility of noisy speech with values from 0.00 to 1.00. A monotonic nonlinear mapping was used to calculate the percentage of correct words based on the STOI findings. Applying a mapping function to the STOI data yields the projected intelligibility scores in this study. The two metrics are:

$$STOI = \frac{10}{1 + exp(cSTOI + d)} \tag{10}$$

$$PESQ = \alpha_0 + \alpha_1 D_1 + \alpha_2 D_2 \tag{11}$$

Where $c$ = -17.49, $d$ = 9.692, $\alpha0$ = 4.5, $\alpha1$ = -0.1, $\alpha3$ = -0.039, $D_1$ denotes the symmetric disturbances while $D_2$ denotes the asymmetric disturbances, respectively.

## Representation of algorithm

Various SE systems are designed with an interpretation indicating the neural network type, with and without skip connections and mask type. (i): LSTM-NoSkips-IRM: This model estimates the IRM training objective by using the proposed LSTM without skip connections. (ii): LSTM-WithSkips-IRM: The model estimates the IRM by using the proposed LSTM with skip connections. (iii): LSTM-AttenSkips-IRM: This model estimates the IRM by using the

proposed LSTM with attention skip connections. The baseline LSTM [16] is represented as LSTM-IRM with IRM as a training target. TIMIT, LibriSpeech, and VoiceBank datasets are used to train all networks.

## Results and discussions

Table 4 presents an evaluation of the proposed SE using STOI in three seen noises. The proposed LSTM model using the combined features and attention skips outscored the networks that are using no skips or using skips with no attention. We observed better STOI (intelligibility) and PESQ (quality) than the counterparts and unprocessed noisy speech with the proposed model. For example, the LSTM-AttenSkips-IRM improved the STOI by 7.7% over unprocessed speech (UNP stands for noisy speech) at -8dB babble noise. Similarly, LSTM-AttenSkips-IRM increased STOI by 23.9% over unprocessed speech at -4dB of car noise. Also, at 0dB factory noise, LSTM-AttenSkips-IRM increased STOI by 20.2% over unprocessed noisy speech. In comparison to the LSTM-WithSkips-IRM, the proposed models with attention skips improved the STOI by 2.1% at -8dB babble noise. Also, the proposed model with attention skips improved the STOI by 9.1% with LSTM-NoSkips-IRM at -8dB babble noise. As a whole, the LSTM-AttenSkips-IRM outperformed and increased average STOI over unprocessed noisy speech as well as SNRs by 1.23%.

Table 5 evaluates the proposed SE models in terms of PESQ for three seen noise types with IRM as an estimated training target. For the PESQ, the suggested LSTM model with combined feature sets and attention skips outscored other models that have no skips or skips with no attention mechanism. We achieved a better perceptual speech quality as compared to the counterparts and noisy speech with the proposed models. For example, in Table 4, the LSTM-AttenSkips-IRM improved the PESQ by 0.34 (20.98%) over unprocessed speech at -8dB factory noise. Similarly, LSTM-AttenSkips-IRM improved the PESQ by 0.54 (26.21%) over unprocessed speech at -4dB babble noise. Moreover, at 0dB car noise, the LSTM-AttenSkips-IRM improved the PESQ by 1.04 (39.1%) over the noisy speech. In contrast to the LSTM-WithSkips-IRM, the proposed models with attention skips improved the PESQ by 0.09 (3.04%) at 4dB car noise. It indicates that at good SNRs (SNR≥4dB) the proposed LSTM model performs almost similarly. In addition, the proposed model with attention skips improved the PESQ by 0.14 (5.28%) with LSTM-NoSkips-IRM at 4dB babble noise. Again, the LSTM-AttenSkips-IRM outscored and increased the average PESQ score over the unprocessed noisy speech as well as SNRs by 3.07%.

**Table 4. STOI scores in seen noise sources for IRM training-target.**

| Noise | Algorithm | -8dB | -4dB | 0dB | 4dB | Average |
|-------|-----------|------|------|-----|-----|---------|
| Babble Noise | Noisy (UNP) | 48.2 | 58.1 | 67.1 | 76.2 | 62.4 |
| | LSTM-NoSkips | 52.7 | 66.7 | 77 | 84.8 | 70.3 |
| | LSTM-WithSkips | 53.8 | 68.8 | 79.1 | 87 | 72.2 |
| | LSTM-AttenSkips | 55.9 | 70.1 | 80.3 | 88.6 | 73.7 |
| Car Noise | Noisy (UNP) | 51.8 | 58.9 | 68.6 | 77.1 | 64.1 |
| | LSTM-NoSkips | 72.4 | 79.2 | 84.9 | 89.4 | 81.5 |
| | LSTM-WithSkips | 74.5 | 86.9 | 86.9 | 91.6 | 85 |
| | LSTM-AttenSkips | 75.7 | 88.3 | 88.3 | 93.2 | 86.4 |
| Factory Noise | Noisy (UNP) | 55.2 | 61.7 | 69.8 | 78 | 66.2 |
| | LSTM-NoSkips | 66.3 | 76.7 | 84.5 | 90.5 | 79.5 |
| | LSTM-WithSkips | 68.3 | 78.8 | 86.5 | 92.6 | 81.6 |
| | LSTM-AttenSkips | 69.5 | 79.9 | 87.9 | 93.7 | 82.8 |

**Table 5. PESQ scores in seen noise sources for IRM training-target.**

| Noise | Algorithm | -8dB | -4dB | 0dB | 4dB | Average |
|---|---|---|---|---|---|---|
| Babble Noise | Noisy (UNP) | 48.2 | 58.1 | 67.1 | 76.2 | 62.4 |
| | LSTM-NoSkips | 1.65 | 1.88 | 2.17 | 2.51 | 2.05 |
| | LSTM-WithSkips | 1.68 | 1.9 | 2.2 | 2.54 | 2.08 |
| | LSTM-AttenSkips | 1.79 | 2.06 | 2.25 | 2.65 | 2.19 |
| Car Noise | Noisy (UNP) | 51.8 | 58.9 | 68.6 | 77.1 | 64.1 |
| | LSTM-NoSkips | 1.97 | 2.26 | 2.57 | 2.84 | 2.41 |
| | LSTM-WithSkips | 2.01 | 2.29 | 2.6 | 2.87 | 2.44 |
| | LSTM-AttenSkips | 2.1 | 2.34 | 2.66 | 2.96 | 2.52 |
| Factory Noise | Noisy (UNP) | 55.2 | 61.7 | 69.8 | 78 | 66.2 |
| | LSTM-NoSkips | 1.52 | 1.86 | 2.16 | 2.53 | 2.01 |
| | LSTM-WithSkips | 1.55 | 1.87 | 2.17 | 2.55 | 2.04 |
| | LSTM-AttenSkips | 1.62 | 2.01 | 2.26 | 2.67 | 2.14 |

The results indicate that LSTM-AttenSkips achieved better PESQ and STOI values. The average PESQ and STOI improvements (PESQi and STOIi) in background noises are depicted in Figs 3 and 4, respectively.

In other sets of experiments, we used the LibriSpeech dataset and Ideal Binary Mask (IBM) to evaluate the proposed SE models. The LibriSpeech is obtained from audiobooks and is composed of 1000 hours of speech sampled at 16 kHz. In experiments, we selected only clean utterances and again mixed them with noise types: airport, babble, street, cafeteria, and car noise at the same SNRs. The average PESQ and STOI values using 5 noises are given in Table 6. The LSTM-WithSkips-IRM and LSTM-WithSkips-IBM have increased the average STOI by 16.44% and 14.9% over unprocessed noisy speech. Further, the LSTM-AttenSkips-IRM and LSTM-AttenSkips-IBM have increased the average PESQ scores with 0.78 (33.19%) and 0.71 (31.14%) over unprocessed noisy speech. We used the VoiceBank dataset to further evaluate the proposed SE models. In experiments, we selected only clean utterances and again mixed them with noise types: airport, babble, street, cafeteria, car, sports field, and well-visited city park noise at the same SNRs. The average STOI and PESQ scores for different noises are given in Table 7. The LSTM-AttenSkips-IRM and LSTM-AttenSkips-IBM have increased the average STOI by 17.21% and 15.4% over unprocessed noisy speech. In addition, LSTM-AttenSkips-IRM and LSTM-AttenSkips-IBM have increased the average PESQ with 0.81 (35.22%) and 0.75 (34.31%) over unprocessed noisy speech.

## Generalization performance

To examine the proposed SE models in terms of generalization, Table 8 provides the PESQ and STOI scores in two unseen noise types (factory2 and cafeteria). The proposed SE models outscored the baseline and the competing networks with significant margins in unseen noises. During analysis, it is observed that the proposed LSTM-WithSkips-IRM and LSTM-WithSkips-IBM obtained the highest 7intelligibility (STOI) and perceptual quality (PESQ) scores since the network architecture is modified to obtain better results. As the suggested models have been treated using robust acoustic feature sets and modifications, their performances are not drastically altered both in unseen or seen noisy conditions. The average STOI values have increased from 63.1% to 78.0% and 76.8% with LSTM-WithSkips-IRM and LSTM-WithSkips-IBM, improving the STOI by 14.9% and 13.7% over unprocessed speech. At low SNRs such as -4dB and -8dB, LSTM-WithSkips-IRM and LSTM-WithSkips-IBM have increased STOI by

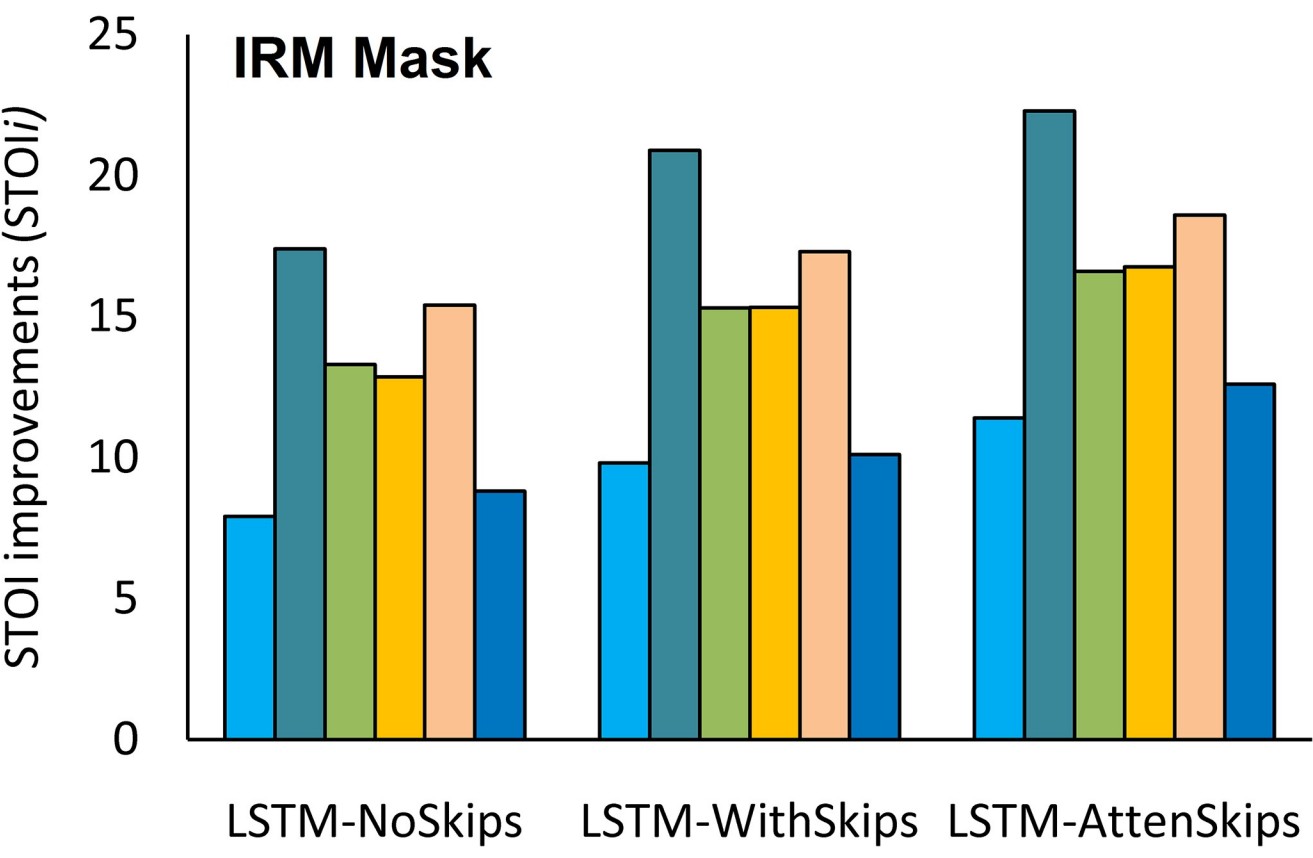

**Fig 3. The STOI improvements (STOIi) in background noises.**

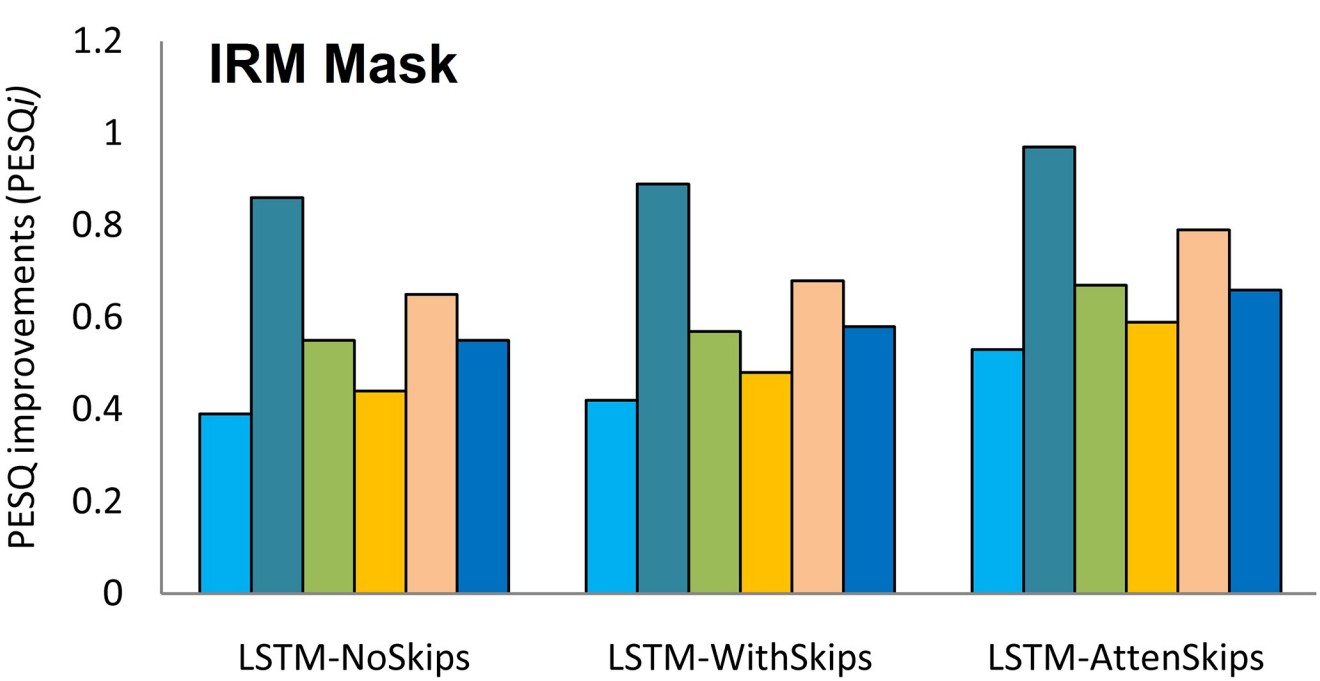

**Fig 4. The PESQ improvements (PESQi) in background noises.**

**Table 6. PESQ and STOI result for LibriSPeech dataset.**

| Metric | Algorithm | Ideal Ratio Mask | | | | Ideal Binary Mask | | | |
|--------|-----------|-------|-------|-------|-------|-------|-------|-------|-------|
| | | -8dB | -4dB | 0dB | 4dB | -8dB | -4dB | 0dB | 4dB |
| STOI | LSTM-NoSkips | 63.9 | 75 | 81.7 | 88.3 | 62.9 | 71.1 | 80.4 | 86.3 |
| | LSTM-WithSkips | 66.1 | 78.6 | 84.3 | 90.2 | 64.4 | 74.7 | 81.4 | 88.2 |
| | LSTM-AttenSkips | 67.4 | 79.7 | 86 | 92 | 66.4 | 77 | 84.9 | 90.6 |
| PESQ | LSTM-NoSkips | 1.72 | 2.04 | 2.3 | 2.66 | 1.66 | 1.98 | 2.28 | 2.61 |
| | LSTM-WithSkips | 1.75 | 2.11 | 2.41 | 2.74 | 1.71 | 2.03 | 2.31 | 2.68 |
| | LSTM-AttenSkips | 1.83 | 2.21 | 2.49 | 2.86 | 1.8 | 2.11 | 2.43 | 2.77 |

**Table 7. PESQ and STOI result for the VoiceBank dataset.**

| Metric | Algorithm | Ideal Ratio Mask | | | | Ideal Binary Mask | | | |
|--------|-----------|-------|-------|-------|-------|-------|-------|-------|-------|
| | | -8dB | -4dB | 0dB | 4dB | -8dB | -4dB | 0dB | 4dB |
| STOI | LSTM-NoSkips | 64.1 | 75.2 | 81.9 | 88.6 | 63.3 | 71.6 | 80.9 | 86.8 |
| | LSTM-WithSkips | 66.3 | 78.8 | 84.6 | 90.5 | 64.9 | 75.0 | 82.0 | 88.7 |
| | LSTM-AttenSkips | 67.7 | 80.0 | 86.3 | 92.8 | 67.0 | 78.1 | 85.3 | 91.0 |
| PESQ | LSTM-NoSkips | 1.76 | 2.10 | 2.36 | 2.72 | 1.69 | 2.01 | 2.31 | 2.63 |
| | LSTM-WithSkips | 1.80 | 2.15 | 2.47 | 2.79 | 1.75 | 2.06 | 2.36 | 2.71 |
| | LSTM-AttenSkips | 1.87 | 2.28 | 2.53 | 2.91 | 1.84 | 2.10 | 2.46 | 2.80 |

1.90% and 1.80% over the baseline LSTMs (LSTM with IRM and LSTM with IBM). Further, the average PESQ values are increased from 1.50 to 2.22 (32.43%) and 2.17 (31.90%) with LSTM-WithSkips-IRM and LSTM-WithSkips-IBM, improving the PESQ significantly over the UNP in unseen noisy conditions. The proposed LSTM models have increased STOI by 1.80% and 2.90% over the baseline LSTMs. The proposed models have increased PESQ by 0.10 (4.54%) and 0.16 (7.27%) over the baseline LSTMs. The proposed models for SE achieved the best performance in unseen noises.

The computational load of the proposed model is measured with trainable parameters and FLOPs (floating-point operations), useful metrics for calculating computational complexity and optimizing the performance on specific hardware platforms. The parameters count and FLOPs for the proposed LSTM model are 26.47M and 127.72 [G] whereas the parameters count and FLOPs for the baseline LSTM are 53.93M and 245.67 [G], respectively indicating the better performance in terms of model complexity and trainable parameters.

## Comparisons with other DL methods

This section examines the performance in terms of average values (STOI and PESQ) obtained by the proposed models and the competing DL models. The experimental results indicate that the proposed LSTM models improved the speech quality, intelligibility, noise suppression, and speech distortion, and also outperformed the baseline LSTM [16], DNN [42], CNN [43], GAN (3-layer ReLU MLP) [44], CNN-GRU [45], and FCNN [46]. Table 8 indicates the generalization capabilities of the suggested LSTM and other DL models. All DL models have been trained using a similar dataset comprising male and female speakers. The experimental values are averaged over all SNRs (-8dB, -4dB, 0dB, and 4dB) and noises. The results in Table 9 indicate that the suggested LSTM models have increased intelligibility and perceptual speech quality. The LSTM-AttenSkips-IRM and LSTM-AttenSkips-IBM have increased STOI by 4.4% and

**Table 8. The quality (PESQ) and intelligibility (STOI) results in unseen noises.**

| Algorithm | STOI | | | | | PESQ | | | | |
|---|---|---|---|---|---|---|---|---|---|---|
| | -8dB | -4dB | 0dB | 4dB | Avg | -8dB | -4dB | 0dB | 4dB | Avg |
| Noisy Speech (UNP) | 50.3 | 58.3 | 67.5 | 76.3 | 63.1 | 1.15 | 1.39 | 1.58 | 1.88 | 1.50 |
| LSTM-AttenSkips-IRM | 64.3 | 74.8 | 82.7 | 90.0 | 78.0 | 1.79 | 2.05 | 2.34 | 2.69 | 2.22 |
| LSTM-AttenSkips-IBM | 63.4 | 72.7 | 82.0 | 88.9 | 77.8 | 1.76 | 1.98 | 2.28 | 2.65 | 2.17 |
| LSTM-IRM (Chan [16]) | 62.0 | 72.9 | 81.3 | 88.2 | 76.1 | 1.62 | 1.91 | 2.24 | 2.64 | 2.10 |
| LSTM-IBM (Chan [16]) | 61.3 | 70.8 | 80.6 | 87.0 | 75.0 | 1.58 | 1.77 | 2.20 | 2.60 | 2.04 |

**Table 9. Comparison of the PESQ and STOI values against competing DL methods.**

| Algorithm | STOI | | | | | PESQ | | | | |
|---|---|---|---|---|---|---|---|---|---|---|
| | -8dB | -4dB | 0dB | 4dB | Avg | -8dB | -4dB | 0dB | 4dB | Avg |
| Noisy Speech (UNP) | 51.7 | 59.6 | 68.5 | 77.1 | 64.2 | 1.24 | 1.48 | 1.67 | 1.91 | 1.58 |
| LSTM-AttenSkips-IRM | 65.5 | 76.0 | 83.9 | 90.3 | 79.0 | 1.81 | 2.09 | 2.39 | 2.72 | 2.25 |
| LSTM-AttenSkips-IBM | 64.6 | 74.9 | 83.2 | 89.1 | 78.0 | 1.77 | 2.03 | 2.35 | 2.67 | 2.21 |
| LSTM-IRM (Chan [16]) | 63.5 | 74.2 | 82.4 | 88.9 | 77.3 | 1.71 | 2.00 | 2.33 | 2.67 | 2.18 |
| LSTM-IBM (Chan [16]) | 62.7 | 72.1 | 81.7 | 87.8 | 76.1 | 1.67 | 1.86 | 2.29 | 2.63 | 2.11 |
| DNN-IRM (Zheng Zhang [42]) | 58.5 | 70.0 | 78.7 | 85.6 | 73.2 | 1.57 | 1.75 | 2.19 | 2.53 | 2.01 |
| DNN-IBM (Zheng Zhang [42]) | 56.1 | 67.3 | 76.5 | 83.1 | 70.8 | 1.49 | 1.70 | 2.11 | 2.45 | 1.94 |
| CNN (Kounovsky and Malek [43]) | 59.3 | 70.0 | 79.8 | 86.8 | 74.0 | 1.62 | 1.83 | 2.25 | 2.59 | 2.07 |
| GAN (Shah et al [44]) | 54.3 | 65.0 | 75.7 | 82.6 | 70.0 | 1.53 | 1.72 | 2.15 | 2.44 | 1.96 |
| CNN-GRU (Hasannezhad et al [45]) | 63.8 | 74.6 | 83.1 | 90.1 | 77.9 | 1.74 | 2.01 | 2.34 | 2.65 | 2.19 |
| FCNN (Ouyang et al [46]) | 60.3 | 71.6 | 79.3 | 86.3 | 74.3 | 1.61 | 1.78 | 2.21 | 2.59 | 2.05 |

6.7% over the DNNs with IRM and IBM as training targets. Further, the LSTM-AttenSkips-IRM has increased STOI by 5.10% (over CNN) and 9.7% (over GAN). Moreover, the LSTM-AttenSkips-IBM has increased STOI by 4.90% (over CNN) and 9.50% (over GAN). Using PESQ, the LSTM-AttenSkips-IRM, and LSTM-AttenSkips-IBM have increased values by 0.20 (9.09%) over CNN and 0.31 (14.09%) over GAN. The overall average improvement of competing models over noisy speech is shown in Fig 5.

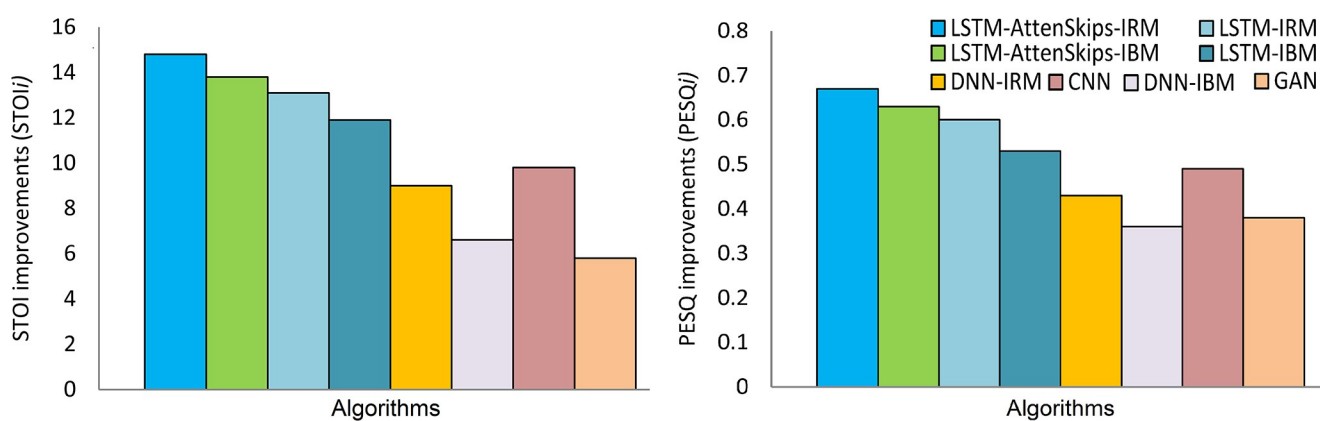

**Fig 5. The average STOI and PESQ improvements (STOIi and PESQi) of deep learning models over noisy speech.**

To visualize spectral regions of the speech processed by deep learning models and the proposed LSTM models, we show spectro-temporal analysis. Fig 6 demonstrates the spectrograms of the utterances. The underlying clean utterance (depicted in Fig 6a) is contaminated at 0dB babble noise in order to create a noisy utterance (depicted in Fig 6b). The babble noise (originated when many people talk simultaneously) is a difficult noisy situation because the noise signal follows the attributes similar to the underlying clean speech. The enhanced speech produced by the LSTM-IBM is illustrated in Fig 6(c), where the background babble noise is considerably eliminated. The enhanced speech produced by the LSTM-IRM (depicted in Fig 6d) shows minimum residual noise and speech distortion in comparison to the LSTM-IBM. Fig 6e illustrates the speech enhanced by the LSTM-AttenSkips-IBM. Minimum speech distortion and residual noise are noticeable. Fig 6f depicts the speech enhanced by the LSTM-AttenSkips-IBM. We can observe that the proposed model reduced the background noise leaving minimal residual noise and speech is not distorted, as confirmed by the spectrogram of noisy speech enhanced by the proposed LSTM-AttenSkips.

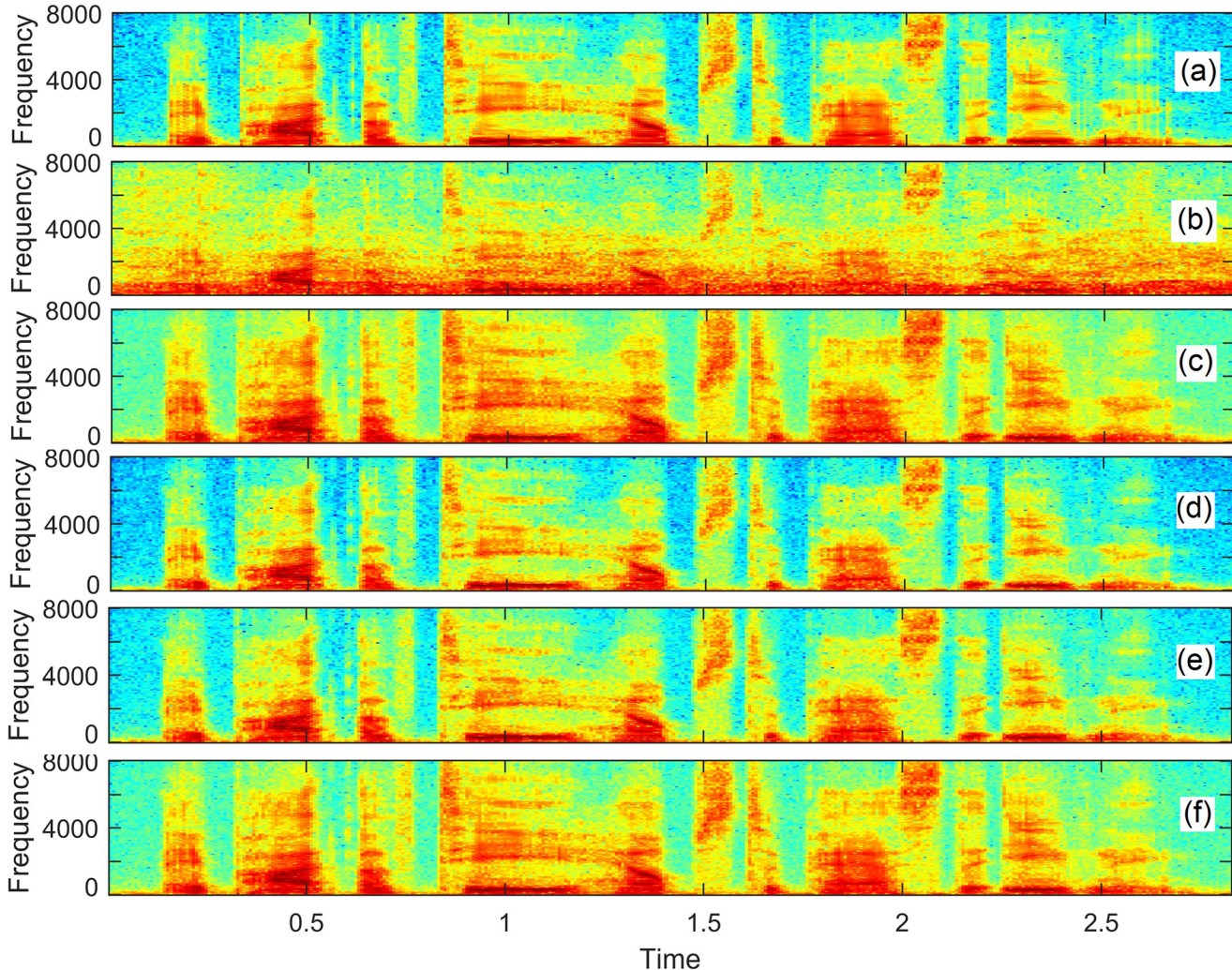

**Fig 6. Visualization of spectral regions.** The underlying clean speech (a), the babble noise-contaminated noisy speech (b), speech processed by LSTM-IBM (c), speech processed by LSTMIRM (d), speech processed by the LSTM-AttenSkips-IBM (e), and speech processed by the LSTMAttenSkips-IRM (f).

**Table 10. ASR performance, where all results are averaged over all SNR levels.**

| LSTM-AttenSkipps-IRM | LSTM-IRM | LSTM-IBM | DNN-IRM | DNN-IBM |
|---|---|---|---|---|
| 15.01 | 20.11 | 21.14 | 28.22 | 28.58 |

## Automatic Speech Recognition (ASR)

The SE evaluations show that the proposed LSTM models greatly suppressed the background noise and recovered high-quality and intelligible speech. As a result, we expect better speech recognition performance in challenging noisy backgrounds. The proposed SE models are implemented at the front end to achieve better ASR results. We implemented the Kaldi toolkit [47] which uses the GMM-HMM system and trained deep neural networks with Mel-frequency filter-bank features. The training system is motivated by Tachioka [48]. We evaluated ASR performance in terms of word error rates (WERs). We randomly selected 2000 speech utterances from the TIMIT and LibriSpeech datasets to train the proposed LSTM-based speech enhancement models. With the trained LSTM models, we performed the speech enhancement and then synthesized time-domain utterances to create new training and testing datasets. We trained ASR models using the new training dataset and tested the ASR models using the new testing dataset. As given in Table 10, the ASR systems when trained with the utterances processed by LSTM-AttenSkips performed better. The WERs gradually decreased with the favorable SNR levels. On average, 19.13% WERs are achieved with the utterances processed by the proposed LSTM-AttenSkips, demonstrating that the proposed SE can be employed as a front-end to boost the ASR performance.

## Conclusion

In this paper, we propose a speech enhancement algorithm that is based on recurrent neural networks trained with robust acoustic feature sets. An hourglass LSTM model is proposed which successfully captures the long-term temporal dependencies by reducing feature resolutions. We used skip connections between the nonadjacent symmetrical layers to prevent the gradient decay over layers. Moreover, an attention mechanism is adopted in skips to highlight the important features and spectral regions. A combined robust feature set is extracted from the magnitude of the noisy speech to robustly train the proposed models for better performance. Two masks, IRM and IBM, are estimated independently. The results have concluded the following aspects of the proposed SE algorithm.

By using the combined features learning, the model includes additional information which enabled the model to better learn the non-linear relation between noisy and clean speech which is confirmed by the results in Tables 4–8. The proposed LSTM models successfully captured long-term temporal dependencies and reduced the feature resolution by using an hourglass architecture to estimate the model parameters for testing which are confirmed by a comparison in Table 8 in the results. The memory overflow is avoided by using the proposed architecture. The skips and attention gate in the skips considerably improved the gradient decay over the layers and also highlighted the important features and spectral regions. The addition of attention gates in the skips obtains better results as indicated by Tables 6 and 7 on two different databases. With the hourglass strategy, the proposed models performed better than the baseline in terms of trainable parameters (18.89M with the proposed and 46.18M with the baseline). The proposed models performed better and outscored the recent deep learning models in different noises as indicated by Table 9. The proposed models also outperformed the related deep-learning methods in unseen noises as confirmed by Table 8. The

Kaldi ASR results demonstrated that the proposed LSTM-AttenSkips SE can be employed as a front-end to boost the ASR performance in noisy backgrounds, confirmed by Table 10 where the proposed model achieves lower WERs.

Phase plays a vital role in improving the perceptual speech quality where a complex Phase spectrum can add significant quality and intelligibility improvements in speech enhancement system [49–51]. The focus of this study is to estimate speech magnitude enhancement where the noisy phase is used during speech waveform reconstruction. Our future study will focus on the simultaneous estimation of the magnitude and phase of the speech and intend to integrate the estimates with the proposed model topology. Further, robust loss functions and feature sets will be developed for improved speech quality and intelligibility.

## Author Contributions

**Conceptualization:** Zhenqing Li.

**Data curation:** Zhenqing Li, Atif Jan.

**Formal analysis:** Atif Jan.

**Funding acquisition:** Abdul Basit.

**Investigation:** Zhenqing Li, Abdul Basit.

**Methodology:** Abdul Basit, Amil Daraz.

**Project administration:** Amil Daraz.

**Software:** Abdul Basit, Amil Daraz.

**Supervision:** Zhenqing Li.

**Validation:** Abdul Basit, Atif Jan.

**Writing – original draft:** Abdul Basit.

**Writing – review & editing:** Amil Daraz.

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
