## [Decision Letter · Decision Letter 0]

14 Jun 2023

PONE-D-23-08666Deep Causal Speech Enhancement and Recognition using

Efficient Long-Short Term Memory Recurrent Neural NetworkPLOS ONE

Dear Dr. Basit,

Thank you for submitting your manuscript to PLOS ONE. After careful consideration, we feel that it has merit but does not fully meet PLOS ONE’s publication criteria as it currently stands. Therefore, we invite you to submit a revised version of the manuscript that addresses the points raised during the review process.

We look forward to receiving your revised manuscript.

Kind regards,

Mohamed Hammad, Ph.D.

Academic Editor

PLOS ONE

Journal Requirements:

2. a) Please note that PLOS ONE has specific guidelines on code sharing for submissions in which author-generated code underpins the findings in the manuscript. In these cases, all author-generated code must be made available without restrictions upon publication of the work. 

Please review our guidelines at https://journals.plos.org/plosone/s/materials-and-software-sharing#loc-sharing-code and ensure that your code is shared in a way that follows best practice and facilitates reproducibility and reuse.

b) Please include a link to the data repository in the Data availbility statement.

"This work is supported by“ Talent Introduction Fund Project of Ningbo Tech University under grant no 20211009."

6. Please update your submission to use the PLOS LaTeX template. The template and more information on our requirements for LaTeX submissions can be found at http://journals.plos.org/plosone/s/latex.

Reviewers' comments:

Reviewer's Responses to Questions

**Comments to the Author**

1. Is the manuscript technically sound, and do the data support the conclusions?

Reviewer #1: Yes

Reviewer #2: Yes

2. Has the statistical analysis been performed appropriately and rigorously? 

Reviewer #1: Yes

Reviewer #2: No

3. Have the authors made all data underlying the findings in their manuscript fully available?

Reviewer #1: Yes

Reviewer #2: Yes

4. Is the manuscript presented in an intelligible fashion and written in standard English?

Reviewer #1: Yes

Reviewer #2: No

5. Review Comments to the Author

Reviewer #1: The following revisions are required.

1. In literature review, add 3 to five more relevant and latest techniques.

2. Add Comparison table at the end of section 2 and compare with at least 7 to 10 techniques with appropriate parameters.

3. Please make sure your paper has necessary language proof-reading.

4. The conclusion is weak and inconsistent with the evidence and arguments.

Reviewer #2: Dear Authors

The paper titled “Deep Causal Speech Enhancement and Recognition using Efficient Long-Short Term Memory Recurrent Neural Networks” proposed an hourglass-shaped LSTM capable of capturing long-term temporal correlations via the reduction of feature resolutions without data loss. The paper is addressing important issues however it needs more improvements.

1. Extensive English editing is required throughout the manuscript.

2. Table 2. The details of the network with hyperparameters of the LSTM and competing Deep Learning Models. These hyperparameters require justification on how they have been chosen and how to tune them.

3. New references need to be updated as very few references are for 2021 and post-2021.

4. Section 1 Introduction: Deep neural networks (DNNs) are effective ….noise conditions over time. This section should have what makes DNN so famous, and which type of recent applications it has to justify its utilization in the present work. I suggest adding smotednn: novel model for air pollution forecasting and aqi classification; cdlstm: a novel model for climate change forecasting; analysis of environmental factors using ai and ml methods; deep learning based modeling of groundwater storage change; deep learning-based supervised image classification using uav images for forest areas classification.

5. Rewrite section 3.4 Representation of Algorithm.

6. # of parameters are required for the models with FLOPS and the computational complexity.

7. Limitations and the future scope should be added with more clarity

6. PLOS authors have the option to publish the peer review history of their article (what does this mean?). If published, this will include your full peer review and any attached files.

Reviewer #1: No

Reviewer #2: No

---

## [Author Response · Author response to Decision Letter 0]

27 Jul 2023

Please see the Word file for reviewers comments.

---

## [Decision Letter · Decision Letter 1]

25 Aug 2023

Deep Causal Speech Enhancement and Recognition using Efficient Long-Short Term Memory Recurrent Neural Network

PONE-D-23-08666R1

Dear Dr. Basit,

We’re pleased to inform you that your manuscript has been judged scientifically suitable for publication and will be formally accepted for publication once it meets all outstanding technical requirements.

Kind regards,

Mohamed Hammad, Ph.D.

Academic Editor

PLOS ONE

Additional Editor Comments (optional):

Reviewers' comments:

Reviewer's Responses to Questions

**Comments to the Author**

1. If the authors have adequately addressed your comments raised in a previous round of review and you feel that this manuscript is now acceptable for publication, you may indicate that here to bypass the “Comments to the Author” section, enter your conflict of interest statement in the “Confidential to Editor” section, and submit your "Accept" recommendation.

Reviewer #2: All comments have been addressed

2. Is the manuscript technically sound, and do the data support the conclusions?

Reviewer #2: Yes

3. Has the statistical analysis been performed appropriately and rigorously? 

Reviewer #2: Yes

4. Have the authors made all data underlying the findings in their manuscript fully available?

Reviewer #2: Yes

5. Is the manuscript presented in an intelligible fashion and written in standard English?

Reviewer #2: Yes

6. Review Comments to the Author

Reviewer #2: All comments haven’t been addressed for the revised manuscript titled Deep Causal Speech Enhancement and Recognition using Efficient Long-Short Term Memory Recurrent Neural Network .

7. PLOS authors have the option to publish the peer review history of their article (what does this mean?). If published, this will include your full peer review and any attached files.

Reviewer #2: No

---

## [Editor Report · Acceptance letter]

6 Oct 2023

PONE-D-23-08666R1 

Deep Causal Speech Enhancement and Recognition using Efficient Long-Short Term Memory Recurrent Neural Network 

Dear Dr. Basit:

I'm pleased to inform you that your manuscript has been deemed suitable for publication in PLOS ONE. Congratulations! Your manuscript is now with our production department. 

Kind regards, 

on behalf of

Dr. Mohamed Hammad 

Academic Editor

PLOS ONE